# Layer-Edge Patterns Exploration and Presentation in Multiplex Networks: From Detail to Overview via Selections and Aggregations

**Xitao Zhang [1], Lingda Wu [1,*], Shaobo Yu [1] and Kang Li [2]**

1   Science and Technology on Complex Electronic System Simulation Laboratory, Space Engineering University, Beijing 101416, China; zxthl0707@163.com (X.Z.); ysb2800@163.com (S.Y.)

2   China Satellite Launch and Tracking Control Center, Beijing 100720, China; xtlikang1987@126.com

*   Correspondence: wld@nudt.edu.cn; Tel.: +86-10-6636-4329

**Abstract:** Multiplex networks have been widely used to describe the multi-type connections of entities in the real world. However, researches for multiplex networks visualization unilaterally focus on the presentation of topological structure, lacking of specific high-level information presentation for quantitative comparison of interlayer structure. Users cannot participate in the exploration and freely choose the layers (or sub-graphs, regions, etc.) of interest for structural comparison. Contraposing the layer-edge patterns visual analysis tasks of multiplex networks, this paper puts forward a novel solution for exploration and analysis that tightly couples topological structure and high-level patterns. It mainly contains a multi-force directed model to realize the balanced layout of nodes in multi-layer topology, as well as two kinds of high-level patterns of which the visual representations are, respectively, designed by a familiar metaphor—that is, the similar pattern representation based on the area-proportional Venn diagrams and the interaction pattern representation based on the directed arrows. Furthermore, views association is implemented through underlying data sharing and multiple interactions which can be used to gain insights through the creation of selections of interest and produce high-level infographic-style overviews simultaneously. The experiments on real-world data demonstrate the support of the proposed method for layer-edge patterns analysis tasks in multiplex networks and the effectiveness for analyzing the multi-layer structure of multiplex networks.

**Keywords:** multiplex networks; interactive visual analysis; force-directed model; simulated annealing algorithm; area-proportional Venn diagrams; selections of interest

## 1. Introduction

Network theory usually describes the real system as a single-layer network (or monoplex network) in which nodes and edges, respectively, represent entities and their connections. However, various systems in real life are often intertwined, such as the multiple types of social relationships among users in multi-social platforms [1]; the combination of the air transport network and the rail transport network [2]; the effect of genes on the synthesis of multiple types of proteins; [3] and so on. The traditional single-layer network fails to represent these semantically different relations and may even lead to a wrong description for the real phenomenon [4]. Multiplex networks (a special kind of multi-layer networks) use different types of edges to represent the different connection relationships between entities, which can effectively describe the heterogeneity of multi-type connections of entities in the real world. Over the last few years, the structural measurement, cascade failure, and propagation process for multiplex networks have increasingly become the research hotspots in the field of network science [5].

The single-layer network visualization technology presents the network data to the user through rich graphical representations, thus achieving the purpose for assisting in understanding the internal structure of the network and mining the valuable information hidden in the network data, and has been successfully applied in engineering. For multiplex networks analysis, in addition to the analysis tasks of the single network layer, such as community composition and the inter-community interaction pattern, the key aspect is the correlation or difference between different layers: What is the difference in communities' structure in two different layers? How does the nodes or edges overlap between two different layers? Answering these questions can be significant for multiplex networks analysis.

There are few studies on the visualization of multiplex networks. References [6,7] study the node layout algorithm for a slice model on the 2.5D view by extending the classical node layout algorithm from a single-layer network to multiplex networks. In order to reduce the visual clutter of the node-link diagram, the edge clustering method for multi-relational networks has been proposed by classifying the bundle edges [8]. Multi-type node attribute measures are presented in the form of annular visualization for comparison between the layers [9]. In reference [10], the global view and the focus layer view are designed in the form of ring, which are used to describe the overlapping edge distribution in two-layer networks.

The above research extends the single-layer network visualization method to multiplex networks and, to a certain extent, achieves the purpose of analyzing the multiplex networks structure and identifying key nodes. However, there are still some problems: Lacking of specific and accurate high-level information presentation, which leads to nonsupport quantitative comparison of interlayer structure; focusing on results presentation, where users can't participate in the analysis process and they can't freely choose the layers (or sub-graphs, regions, etc.) of interest for structural comparison, which limits the in-depth exploration of multi-layer structure.

In order to make it convenient for users to freely explore the multi-layer structure features in multiplex networks and analyze the similarity (or difference) between network layers, a novel solution for exploration and analysis is proposed. More specifically, the main contributions of this article are:

- An interactive exploration and analysis model that tightly couples topological structure and high-level patterns;
- a multi-force directed method specially for multiplex networks visualization, which realizes the balanced layout of nodes in multi-layer topology by considering the attraction from the center of the community and the cross-layer attraction from the counterpart node, and then the similar communities between layers can be identified quickly;
- two kinds of high-level patterns, of which the visual representations are, respectively, designed by a metaphor familiar to users—that is, the similar pattern representation based on the area-proportional Venn diagrams and the interaction pattern representation based on the directed arrow, which are convenient for non-expert users to obtain the abstract of interested regions;
- views association, enabling users to gain insights through the creation of selection of interest (layer, sub-graph, region, etc.), and produce high-level infographic-style views simultaneously.

## 2. Related Work

In this section, some kinds of visualization technology are introduced which are related to the implementations in this paper.

Node-link diagrams are always used in traditional network visualization [11]. This representation does well in showing topology of a simple graph but does not meet the need for multi-edge presentation in multiplex networks visualization. In addition, it is difficult for users to perceive real network structure in the topology view with "hairball-like" visualizations, as the scale of nodes and edges increases. Based on the implementations of some classical force-directed layout methods [12–14], this paper proposes a multi-force directed layout algorithm and designs a node layout algorithm set which is convenient for users to generate varieties of topological views as needed.

There are two main ways to prevent visual clutters in node-link diagrams: Top-down [15] and bottom-up [16] exploration. In the top-down approach, users start by exploring the entire network overview, where the features of interest can be identified, and continue to explore the substructures of concern. However, this method is difficult to apply to the exploration of large-scale node-link diagrams, because of the difficulty of finding useful features due to a clutter visualization and overload information. Instead, a bottom-up exploration can be initiated with a (predetermined) single node of interest and the neighboring nodes, afterwards. Nevertheless, users are always ambivalent about finding nodes of interest in a chaotic overlapping topology. This paper adopts a hybrid exploration method to achieve the gradual progress from detail to overview in multiplex networks by switching and continuously updating the regions of interest, respectively, in topological structure views and high-level infographic-style views.

Venn diagrams have been widely used in the field of chromosome analysis to reveal the overlaps of gene sequences [17–19]. In Venn diagrams, circles are used to represent sets, the overlap area of the circles represents the intersection of the sets (i.e., repeating elements), and the size of the area represents the potential of the set (i.e., the number of repeating elements). Reference [20] extends the force-directed layout method to calculate the positions of the circles, realizing the automatic drawing of the Venn diagrams. To calculate size of the intersection among the circles, a polygon-based approximation method [21] and a binary index-based statistical method [22] are, respectively, proposed. However, the former is suitable for Bayesian inference analysis but not for information visualization with high accuracy because of the large error in approximate treatment. The latter requires complex calculation for overlap areas in the process of optimizing layout, resulting in large computational cost. As such, it is not appropriate for interactive visual analysis. In this paper, we refer to reference [19] for the representation of repeated gene sequences, and define the first kind of layer-edge pattern, i.e., similarity pattern, and design its visual representation for multiplex networks. Furthermore, a method for directly calculating the size of intersection and the optimal layout method are applied to reduce the complexity in circles layout.

In movement data analysis, a "motion pattern" is used to visualize and explore the flow density distribution and mobility pattern over time and space [23]. In reference [24], each ridership is represented as a curved ribbon, showing the passenger travel pattern from multiple spatial scales and multiple time scales, based on the interactive circos diagrams. In reference [25], the interchange pattern is represented as a directed arrow and the multi-dimensional attribute analysis is carried out on the immigration data. Making uses for reference the concept of interchange pattern from movement data analysis, this paper defines the second kind of layer-edge pattern, i.e., interaction pattern, and designs its visual representation as a directed arrow to describe the connection between regions (or inside each region self) for a single layer of multiplex networks.

## 3. Theory and Methods

### 3.1. Model of Multiplex Networks

In this section, we will briefly introduce the model of multiplex networks and some related definitions, which are the basic theories for task analyzing and visual presentation designing in multiplex networks visual analysis.

A multiplex network is a generic data structure made of multiple layers, where each layer can be treated as a mono-plex network (i.e., network with a single layer). In addition, the same node can appear in multiple layers and nodes can be connected only in the same layer. As such, a multiplex network $G = \{L_i; i \in \{1, \cdots, d\}\}$ can be described as a family of (directed or undirected, weighted, or unweighted) graphs $L_i = (V_i, E_i)$ [5], where $d$ is the number of all possible layers, $V_1 = V_2 = \cdots = V_d = V$ is the collection of all nodes, and $E_i \subseteq V_i \times V_i$ is the set of edges that connected pairs of nodes in the layer $L_i$. In a multiplex network, an identical set of nodes can be connected by different types of links, and each type of links is represented as a layer.

**Definition 1.** *The aggregation layer is a special layer which represents the topology of the entire multiplex network in a single network layer. As such, the aggregation layer can be described as $(V_A, E_A)$, where $V_A = V$ is the collection of all nodes, $E_A = \overset{d}{\underset{i=1}{\cup}} E_i$ is the union of all edges in every layer; that is, if a pair nodes are connected in any layer, this pair of nodes have an edge in the aggregation layer.*

**Definition 2.** *The overlap layer is a substructure of any layer in which each node pair in the overlapping layer is connected if and only if the corresponding pair of nodes in all layers is connected. The overlap layer can be described as $(V_O, E_O)$, where $V_O = V$ is the collection of all nodes, $E_O = \overset{d}{\underset{i=1}{\coprod}} E_i$ is the collection of edges which appears in every layer. In other words, the overlap layer can be seen as the core structure of the multiplex network.*

Figure 1 shows the schematic diagrams of multiplex networks model. Sub-figure (**a**) shows a multiplex network with three layers, *L1*, *L2*, *L3*; each layer contains seven nodes; there are a total of 17 edges in this multiplex network. Figure 1b shows the aggregation layer of this multiplex network, where all the nodes and edges are represented in a single network layer and the number of edges in this layer is nine. Figure 1c is the overlap layer, in which the edge is represented that only appears in each layer, so the total number of edges in the overlap layer is 3.

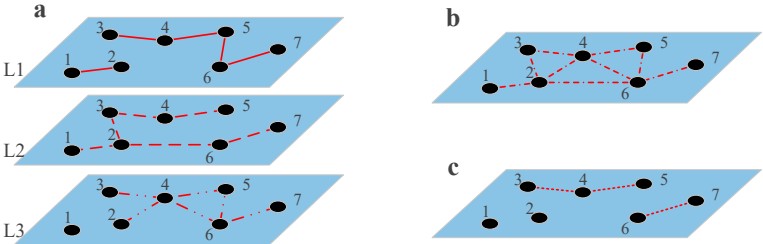

**Figure 1.** Multiplex networks model: (**a**) Multi-layers, (**b**) aggregation layer, and (**c**) overlap layer.

*3.2. Tasks for Multiplex Networks Visual Analysis*

In the field of multiplex networks analysis, different types of edges represent different interactions. One of the key aspects of multi-layer structure analysis is the similarity or difference of topology between different layers: What is the difference in the composition of different layers in terms of communities? Is there a similar community between two or more layers? How do they overlap in terms of nodes or edges? Answering these questions can be very helpful for multiplex network topology analysis.

Through a summary of large numbers of studies on the taxonomy of network visualization tasks, Brehmer and Munzner [26] indicated that the main disadvantage of most methods is the lacking of a global perspective on the taxonomy. In other words, the high-level taxonomy usually ignores the execution mode of tasks, while the low-level categories usually ignore the reasons for performing of the tasks. To compensate for this gap, Saket et al. [27] studied a multi-level taxonomy for single-layer network visualization tasks. This approach helps to create a complete task description that divides the tasks into four subcategories: Group tasks, group-node tasks, group-link tasks, and group-network tasks.

According to the taxonomy described above and features of multiplex networks [6–10], we propose a specialized visual analysis task set for layer-edge patterns exploration in multiplex networks by expanding the "layer level" tasks. The tasks can be divided into two types: Single-layer analysis and multi-layer analysis. Furthermore, each type of tasks can be refined in accordance with three levels, i.e., layer, group, node (edge). The specific tasks are shown in Table 1.

**Table 1.** List of tasks of layer-edge patterns analysis in multiplex networks.

| Task 1: Single-Layer Analysis |
| --- |
| Task 1.1: Structure component in terms of group of a given layer (Layer-Group Level) |
|     Task 1.1.1: Number and distribution of nodes in groups of a given layer (Group-Node Level) |
|     Task 1.1.2: Number and distribution of edges in groups of a given layer (Group -Edge Level) |
| Task 1.2: Number of nodes of a given layer (Layer-Node Level) |
| Task 1.3: Number of edges of a given layer (Layer-Edge Level) |
| **Task 2: Multi-Layer Analysis** |
| Task 2.1: Comparison between two or more given layers (Layer Level) |
|     Task 2.1.1: Comparison in terms of group between two or more given layers (Layer-Group Level) |
|     Task 2.1.2: Comparison in terms of the number and distribution of overlap nodes between two or more given layer (Layer-Node Level) |
|     Task 2.1.3: Comparison in terms of the number and distribution of overlap edges between two or more given layer (Layer-Edge Level) |
| Task 2.2: Comparison in given groups between two or more given layers (Group Level) |
|     Task 2.2.1: Comparison in terms of number and distribution of overlap nodes in given groups between two or more given layers (Group-Node Level) |
|     Task 2.2.2: Comparison in terms of number and distribution of overlap edges in given groups between two or more given layers (Group-Edge Level) |

Based on the taxonomy above, this paper defines two high-level information patterns for multiplex networks layer-edge patterns exploration.

**Definition 3.** *The similarity pattern is the description of the number and the distribution of similar communities, overlap nodes, or overlap edges in given layers or groups.*

**Definition 4.** *The interaction pattern is the description of the number and the distribution of edges between given groups in a single layer.*

According to the similarity pattern and interaction pattern, this paper designs three visual representations to support the analysis of the above tasks, respectively: The topological structure representation based on node-link diagrams, the similarity pattern representation based on Venn diagrams, and the interaction pattern representation based on directed arrows.

*3.3. Analysis Model from Detial to Overview*

There are many inefficiencies and inapplicability in the techniques of rendering, user visual perception and exploration, while only node-link diagrams are used to show the details of the network topology. First, large-scale network exploration has been a major challenge because of the size limitations of the canvas and the inability to display dense nodes and edges in the network topology view. Second, for expert users, low-level visualization methods that focus on displaying all individual elements in a view may overload information, making it difficult for analysts to find and focus on areas (or elements) of interest. Again, for non-expert users, they are more likely to get a network overview and useful comparisons through familiar metaphors rather than cluttered details. Furthermore, the areas (or elements) that the user pays attention to are often different. Rich interactions can fully utilize the domain knowledge of the users, can be used to carry out more in-depth and comprehensive exploration, and can be used to obtain more valuable information and rules.

In summary, in order to meet the requirements of interactive exploration of large-scale network data by multiple types of users, we need:

- Multi-style interaction methods—the user can directly operate the visual element and select the layer (or group) of interest;
- the ability to view detailed information and aggregated high-level patterns simultaneously with a familiar metaphor.

Here, we introduce a top-down and bottom-up hybrid exploration approach. The schematic model of the exploration approach is shown in Figure 2.

The topological structure view (see Section 3.4) presents the topology of the network in the form of node-link diagrams, which can clearly and intuitively reflect the network composition and state, as well as assisting users in evaluating network nodes, links, and community components, making it easy for users to select areas and elements of interest.

The analyst selects the elements inside the region of interest by operating on the topology view and dataset through interactions such as selection and filtering (see Section 3.6).

The high-level infographic-style view aggregates the information of nodes and edges inside the selected area and generates one of the two kinds of the representations of the high-level patterns, i.e., a similarity pattern representation (see Section 3.5.1) to show the comparison of interlayer similarities or an interaction pattern representation (see Section 3.5.2) to demonstrate the interaction pattern.

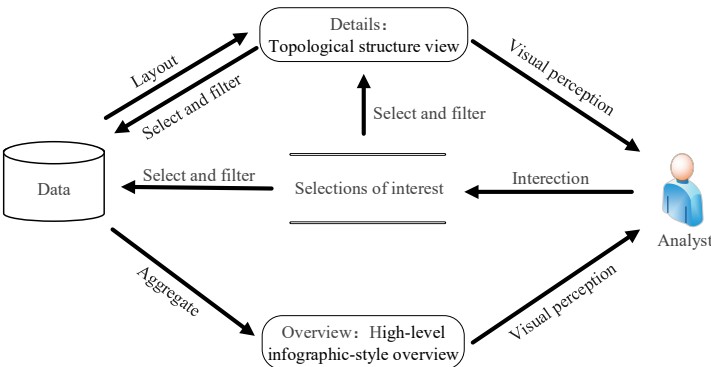

**Figure 2.** Diagram of analysis model from detail to overview based on selection and aggregation.

### 3.4. Topological Structure View

In order to provide a network overview, each node and each edge are displayed in the topology view based on the node-link diagrams, as shown in Figure 3a. The position of the node in the canvas is calculated by the force-directed layout algorithm.

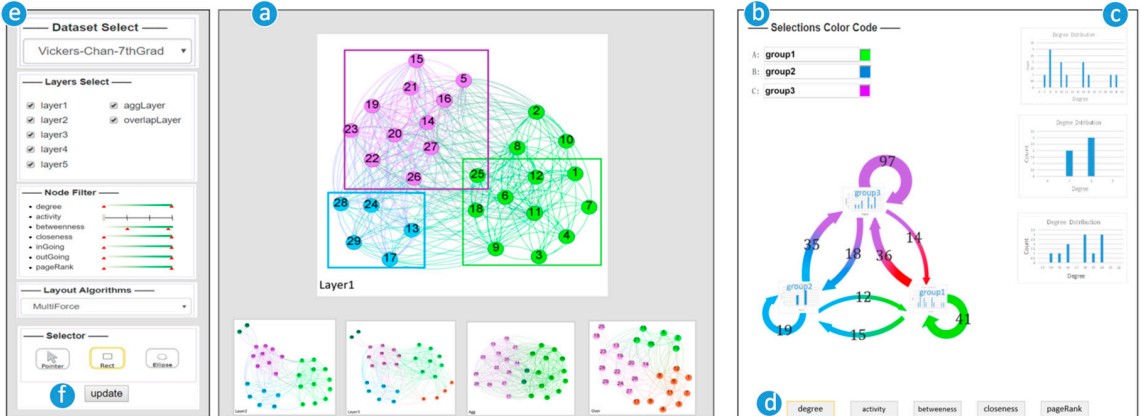

**Figure 3.** Graphical user interface of the implemented prototype showing all coherent components: (**a**) Topological structure view showing all the network layers in 2D panel as node-link diagrams. The layer1 is selected and positioned at the view's center. (**b**) High-level informatic-style view showing the high-level patterns representations with Venn diagrams or directed arrows; the latter is shown here to present the interaction pattern between the three areas are selected in topological structure view. (**c**) Attributes charts views showing the statistical information of nodes in the selected area and which kind is to be shown is controlled by the (**d**) attribute selection component. (**e**) Interaction components showing the operations method to interact with users, respectively: The data inputting, layer selection, node filter, layout algorithms selection, and region selection. (**f**) Update button.

This section mainly introduces a multi-force directed model and a node automatic layout method for multiplex networks visualization: First, the Louvain algorithm is used to divide the nodes of each layer into communities; then, a multi-force directed model is built by considering the community's centroid attraction and cross-layer attraction from the counterpart node; finally, the simulated annealing algorithm is introduced to add temperature control parameters for the multi-force directed layout, controlling the moving speed of the nodes, and complete the node position calculation.

3.4.1. Community Detection

The Louvain algorithm [28] is a condensing algorithm based on modularity maximization which has been widely used in community detection. Several modified versions of the Louvain algorithm have been proposed which are directed to the improvement of time efficiency [29] or the quality of the result [30]. According to problem of the resolution limit [31], a generalized version of the Louvain algorithm has been proposed in [32,33]. The traditional Louvain algorithm will be introduced as a representative community detection algorithm in this section. What should be noted is that other community detection methods can also be applied to explore the mesoscale structure of the network from different views.

The Louvain algorithm is mainly divided into two phases which are shown in Figure 4. First, each node is initialized as a community, and all the nodes in the network are traversed continuously and taken out from the original community. The modularity increment generated by the points added to each community is calculated. If the modularity increment is greater than zero, the community with the largest increment corresponding to the module is selected, and the point is merged with the point. Repeat the above process until the communities in the network are no longer merged. Then, using the first layer community obtained to construct a new network, and the weight between the new nodes is the weight between the original communities. Repeat the process of the previous phase until the communities can no longer merge.

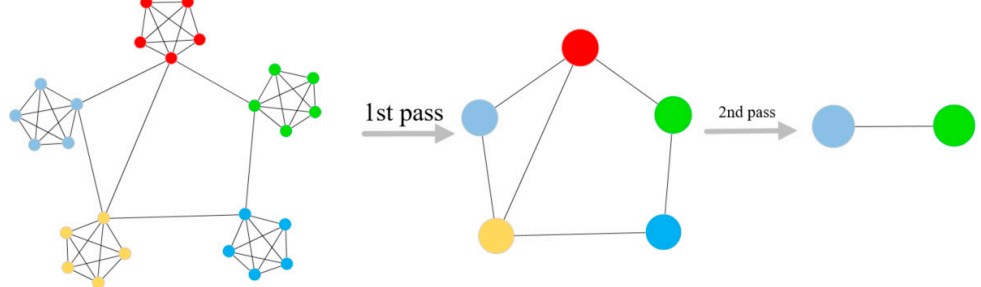

**Figure 4.** Schematic diagram of multi-level community division based on the Louvain algorithm [28].

3.4.2. Multi-Force Directed Model

The attraction force and repulsive force of one node can be, respectively, calculated in the traditional force-repulsion (FR) model as follows:

$$F_a = K_a(D - Length) \tag{1}$$

$$F_r = -K_r/(D * D) \tag{2}$$

where $D$ is the Euclidean distance between nodes, *Length* is the original length of the spring (i.e., the ideal length of the edge), $K_a$ is the spring force constant, and $K_r$ is the repulsion constant According to the principle of "uniform layout of nodes", the ideal length of the edge can be calculated by the following formula:

$$Length = \sqrt{WH/|V|} \tag{3}$$

where $W$ and $H$ represent the width and height of the drawing area, respectively, and $|V|$ is the number of nodes.

As shown in Figure 5, the multi-force directed model is based on the traditional FR model, where the attractive force of node $v$ in the layer $L_a$ consists of three parts: The attractive force between nodes with an intra-layer edge, $F_{a\_intra}(v, L_\alpha)$, the community's centroid attraction $F_{\alpha\_cent}(v, L_\alpha)$, and cross-layer attraction from the counterpart node $F_{a\_inter}(v, L_\alpha)$. As such, the resultant force of the node $v$ in the layer $L_a$ can be calculated as follows:

$$
\begin{cases}
F(v, L_\alpha) = F_r(v, L_\alpha) + F_a(v, L_\alpha) \\
F_r(v, L_\alpha) = \sum_{\mu \in L_\alpha} -K_r/(\Delta * \Delta) \\
F_a(v, L_\alpha) = F_{a\_intra}(v, L_\alpha) + F_{a\_inter}(v, L_\alpha) + F_{cent}(v, L_\alpha) \\
F_{a\_intra}(v, L_\alpha) = \sum_{e \in E(v, L_\alpha)} K_{a\_intra}(\Delta - Length) \\
F_{a\_inter}(v, L_\alpha) = \sum_{L_\beta \in L(v)} K_{a\_inter}(\Delta - Length) \\
F_{\alpha\_cent}(v, L_\alpha) = K_{\alpha\_cent}/(\Delta * \Delta) \\
\Delta = sqrt[(v.x - \mu.x)^2 + (v.y - \mu.y)^2]
\end{cases}
\tag{4}
$$

where $E(v, L_a)$ represents the collection of all the intra-layer edges connected to node $v$ in the layer $L_a$. Since this model uses 2D to display multi-layer multiplex network nodes, $\Delta$ is defined as the horizontal distance between two nodes, $L(v)$ is a collection of the layers where node $v$ appears, and $K_{a\_intra}$, $K_{a\_inter}$, and $K_{a\_cent}$ are, respectively, the intra-layer attraction constant, the inter-layer attraction constant, and the centroid attraction constant.

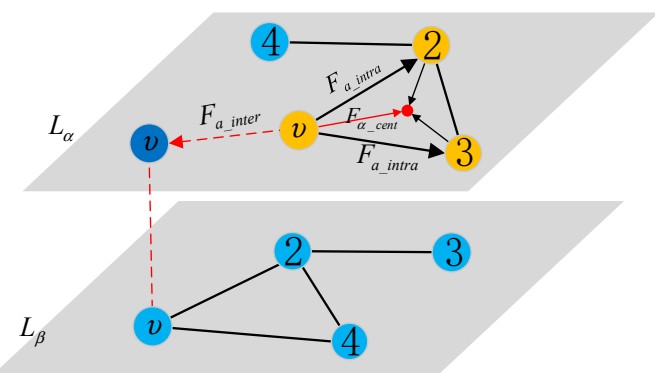

**Figure 5.** Schematic diagram of multi-force directed model.

3.4.3. Iterative Algorithm Based on Simulated Annealing

In order to achieve stable layout quickly and avoid excessive iteration of the layout process, the temperature control parameter *alpha* is set for the multi-force directed layout, according to the simulated annealing idea. The layout algorithm adjusts the velocity of the node during the iterative process by temperature decay: The node moves fast while the temperature is high, and otherwise, the node moves slowly. When the temperature decays to the lower threshold, the velocity is 0, the node layout enters a steady state, and the simulated stress process ends.

The temperature of the system at any time equals to the temperature of the previous moment minus the attenuation value, i.e.,

$$alpha(t + 1) = alpha(t) - alpha(t) * alphaDecay \tag{5}$$

$$alphaDecay = 1 - alphaMin^{1/Iterations} \tag{6}$$

where *alphaDecay* is the temperature decay rate, and *alphaMin* is the lower threshold of system temperature. When the temperature decays to this threshold, the layout is stable. *Iterations* is the number of iterations.

$F(t)$ is the stress of a particle at time $t$. Considering $m = 1$ and $\Delta t = 1$, the resulting velocity increment is

$$\Delta v(t) = F(t)/m * \Delta t = F(t). \tag{7}$$

The node's velocity $v$ is jointly controlled by the system temperature $t$ and the velocity attenuation factor *velocityDecay*. The temperature *alpha* controls the iterative process, and *velocityDecay* controls the velocity attenuation so that the system can stably converge. The velocity at the next moment $v(t+1)$ is

$$v(t+1) = (v(t) + \Delta v(t) * alpha(t)) * velocityDecay. \tag{8}$$

The displacement that the node is generating at $\Delta t$ is

$$\begin{cases} dx_t = v_x(t) * \Delta_t \\ dy_t = v_y(t) * \Delta_t \end{cases}. \tag{9}$$

The node's position is updated at the next moment by the formula:

$$(x, y)_{\{t+1\}} = (x, y)_t + (dx, dy)_t. \tag{10}$$

### 3.5. High-Level Infographic-Style View

As shown in Figure 3b, the high-level infographic-style view represents the aggregated information of the relevant layers, nodes, and edges by the inter-layer similar pattern representation and the intra-layer interaction pattern representation. When a user selects a layer of interest (or sub-graphs, regions, etc.) by multiple interactions or creates a new selection box in the topology view, the system semi-automatically creates a high-level infographic-style view, providing the abstract and insightful aggregated information to users.

### 3.5.1. Similarity Pattern Representation

Nodes and edges repeated in multiple network layers have key supporting roles for multiplex networks functions. This section combines high-level patterns extraction with visualization methods by using area-proportional Venn diagrams to represent the distribution of overlapping nodes (or edges) between multiple network layers and supporting the similarity analysis of topologies of multiple network layers. In the similarity pattern representation, each network layer is treated as a set; nodes (or edges) in the layer are treated as elements in the sets, and the nodes (or edges) overlapping between layers are the intersection of the sets.

The feature of area-proportional Venn diagrams is that the size of each area is proportional to the potential of the set (or intersection). As such, the calculation of the size of the overlap area and the layout of the set position are the two key points to the drawing of the Venn diagrams.

This section introduces a simple method for directly calculating the size of the overlap area. Then, the root mean square error function of the areas is used to evaluate the difference between the current layout and the ideal layout, and the Nelder-Mead method to find the optimal set layout position.

As we know, the intersection area can always be divided into a convex N-gon and a series of arcs areas, no matter how many rings there are. Therefore, the problem of calculating the overlapping size of the N rings can be converted into the problem of finding the sum of the sizes of a convex N-gon and N arc areas.

As shown in Figure 6, the intersection area of the three rings shown in the Figure 6a is divided into a triangle and three arc areas in the Figure 6b. In summary, the calculation formula for the overlapping area of $N$ rings can be expressed as:

$$S_{N-overlap} = S_{N-polygon} + \sum_{i=1}^{N} S_{arc}^{i}. \tag{11}$$

Suppose that the length of each side of an arbitrary triangle is a, b, c, and the calculation formula of the size of the triangle is:

$$S_{\Delta} = \sqrt{p(p-a)(p-b)(p-c)} \tag{12}$$

where $p = \frac{1}{2}(a+b+c)$.

As shown in the Figure 6, the ring radius $r$ and the arc angle are known. The size of the arc area in sub-figure (c) can be obtained by calculating the difference between the fan area and the triangle area, namely:

$$S_{arc} = S_{sector} - S_{\Delta OPQ} \tag{13}$$

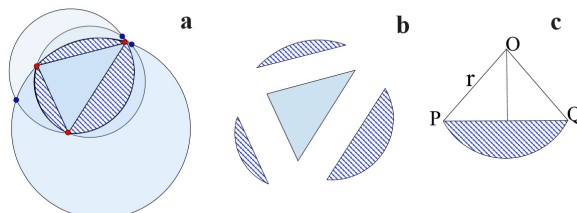

**Figure 6.** Schematic diagram of size calculation of overlap area among rings, (**a**) intersection area of the three rings; (**b**) division of overlapping area; (**c**) diagram of the calculation of arc area.

Here, the calculation steps of the overlapping areas of N rings are given as follows without loss of generality:

Step1:   Count the coordinates of the intersection of N rings and sort them clockwise to determine the centroid coordinates of the overlapping polygons $O_{N-polygon}(x, y) = \frac{1}{N} \sum_{i=1}^{N} (x_i, y_i)$.

Step2:   Calculate the size of the convex N-gon, $S_{N-polygon}$, i.e., the sum of the sizes of N triangles by the formula (12) according to the centroid coordinates and the coordinates of two adjacent intersection points.

Step3:   Calculate the size of each arc region with the formula (13) by selecting the angles between two adjacent intersection points. Then the sum of the sizes of the N arc regions $\sum_{i}^{N} S_{arc}^{i}$ can be calculated.

Step4:   Calculate the sizes of N ring overlap areas by Formula (11).

The above method to calculate the size of the overlap area is more accurate than the method based on polygon approximation proposed in reference [21] and is simpler and more direct than the binary index method proposed in reference [22].

The purpose of the ring layout is to lay out all the rings in ideal places, ensuring that the sizes of all intersections are proportional to the potentials of the intersections of the input sets. Here, we use the root mean square error function of the area to evaluate the current layout effect:

$$Re_S = \sqrt{\sum_{i}^{M} (targetArea_i - currentArea_i)^2} \tag{14}$$

where *targetArea$_i$* and *currentArea$_i$*, respectively, represent the ideal size and current size of the area, and *M* is the number of sets input.

In this paper, the Nelder-Mead method [34] is used to solve the above nonlinear optimization problem. When the minimum value is taken, the best layout can be gotten. The specific algorithm is based on fminsearchbnd () in matlab.

As shown in Figure 7, the relationship between the edges sets of the three network layers is visualized using area-proportional Venn diagrams. From the Figure 7, there are three network layers A, B, and C; the size of C is the largest, indicating that layer C contains the most edges. The size of the intersection among the three sets is 18, indicating that the number of edges shared in three network layers is 18, which constitutes the core structure of the three network layers; the edges in layer B are mostly the overlapping edges with layer A or layer C, and there are fewer edges of its own—in other words, most of the interaction in layer B can be achieved through the other two layers, i.e., layer A and layer C.

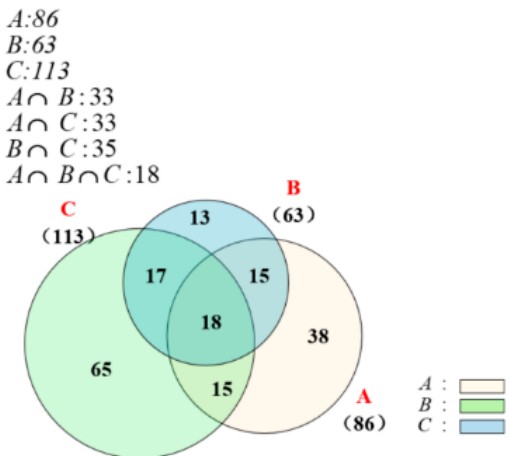

**Figure 7.** Case of area-proportional Venn diagrams.

### 3.5.2. Interaction Pattern Representation

The interaction pattern indicates the aggregation of the detailed information internal of the selected area in a single network layer including some statistical information of the inner edges of the area, the edges between the areas, and the number of nodes.

As shown in Figure 3a, three different areas are selected by using the rectangular frame selection method. The objects selected are all the nodes inside the area. The rectangles are coded in the same color with the corresponding areas. Figure 3b shows the interaction pattern representation between the areas, which consists of a box container and a directed arrow:

1.  Each box container represents a selected area which is coded in the same color as the selection rectangular. Inside of the container, as shown in Figure 3b, is the node degree distribution of each area. The user can click on the box container to zoom in and display the internal information of the container. The histogram of the degree distribution of the internal nodes in each area is shown in Figure 3c. The containers can also display information such as node overlap degree, node activation degree, and node importance ranking in other forms of bubble chart, histogram, or ring chart.

2.  All the intra-edges of the area are displayed as a self-circulating directed arrow; all the inter-edges are shown as a directed arrow between the box containers and coded with a gradient of the colors of the starting area and the target area. The width of the arrow is proportional to the sum of the number of edges associated with the selected area.

An interaction pattern representation of the region of interest can be displayed when a region or regions of interest is selected in the topology view by creating a new selection box on a layer or filtering operations on nodes.

Furthermore, a standard component for selections editing is used to freely set colors and names for different interaction pattern representations based on semantics.

### 3.6. Interaction and Views Association

Interaction is the key to connecting users, data, and visual representations in visual analysis and is the primary mean for views association analysis. When users focus on the parts of interest of the network through a variety of selection operations in the exploration of multiplex networks, as shown in Figure 3d, the details are aggregated in the backend and the accurate pattern information can be directly presented to the user in a simple and intuitive high-level infographic-style diagrams.

#### 3.6.1. Sharing of Underlying Data

In order to support the diverse needs in user's exploration and multi-views contrastive analysis, the underlying data is shared among the multiple views. In other words, the underlying data structure in all views remains exactly same, with the only difference that the ways in which the data is encoded in the different views. As shown in Figure 8, an identical data set is represented in three views. The node-link diagrams in Figure 8a show the nodes and their connections, where the nodes are divided into 2 groups; the histograms in Figure 8b show the degrees of different nodes; and the directed arrows in Figure 8c shows the degree of interaction between the two groups.

Through the sharing of underlying data, the connection relationships, attribute statistics information, and patterns information of the same data set are displayed in the parallel views, ensuring the visual can be switched between multiple views that are simultaneously visible. Thereby, this method produces much lower cognitive load compared with the use of query memory to compare the current view with the last view [31].

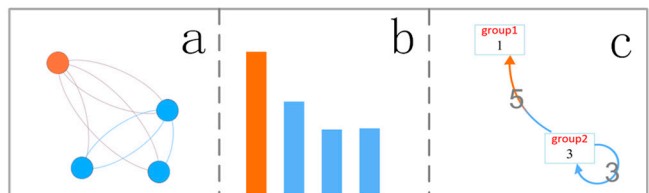

**Figure 8.** Sharing of underlying data for multi-view association analysis, (**a**) node-link diagrams; (**b**) histograms of degree; (**c**) interaction pattern.

#### 3.6.2. Filtering by Node Attribute

There are too many elements in a dense network to distinguish clearly. As such, filtering is quite necessary for removing redundant information and performing in-depth analysis. Node filtering makes it easy to remove the data that is not of interest at the moment, and users can focus on the rest of the data. In addition, filtering doesn't actually delete the data—it just hides the data visually. All data can be made visible again by adjusting the filter's value or closing the filter.

As shown in Figure 9, appropriate nodes and their edges can be filtered by adjusting the filter conditions, while the nodes that do not meet the conditions will be hidden. Then, users can focus on the reserved nodes, proactively explore different parts of the network, and try to understand the associated patterns.

Both the global and local patterns are useful for data analysis. As such, it is convenient for users to efficiently and comprehensively mine the pattern information hidden in the data by switching the region of interest during the exploration process.

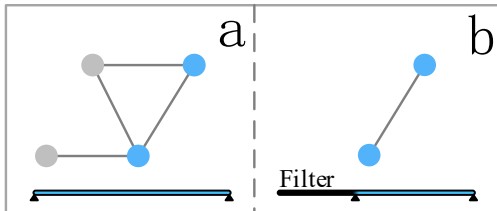

**Figure 9.** Diagrams of filtering interaction model based on node attributes, (**a**) nodes before filtering; (**b**) nodes after filtering.

As shown in Figure 10a, a user creates an area of interest by rectangle selection in the topology view. Then, a high-level information representation of the relevant region is generated in the high-level infographic-style view, just as shown in Figure 10b,c. A selection is all the nodes and their edges that are contained in the area, and the user can change the range of the selection by dragging the box or changing the size of the rectangle.

The area selection operation supports selecting in both the same network layer and cross-layer. The former is used for the interaction pattern analysis between the regions of interest in the same layer; the latter can be used to achieve the similarity comparison of regions of interest in different layers.

Furthermore, a selection sorting mechanism is defined to resolve conflicts between the selections: One node only belongs to one selection; if multiple areas have an overlapping node, the selection the node belongs to is determined according to the selections' order—that is, the node belongs to the selection with the highest ranking.

Other interactive components that support view association analysis include layer selection component and selection setting component.

In addition, rich interactive operation options can make full use of human domain knowledge and multiple coding methods to quickly analyze the network. In other words, users can freely explore and then discover more valuable information.

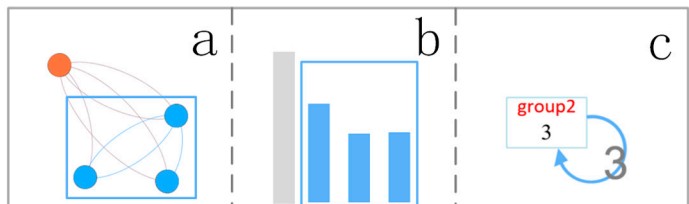

**Figure 10.** Diagrams of selection of interest region, (**a**) node-link diagrams; (**b**) histograms of degree; (**c**) interaction pattern.

## 4. Case Study

In this section, the proposed method is applied to the exploration process on a real multiplex network data set. The support for layer-edge patterns tasks in Section 3.2 is demonstrated to verify the feasibility and effectiveness of the method described above.

### 4.1. Multiplex Networks Data

The data set [35] used in this case was collected by Vickers from 29 seventh grade students in a school in Victoria, Australia. The three layers, respectively, represent different relations between the students, i.e., "get on with,"" best friends," and "prefer to work with". There are 29 nodes and 740 edges in all layers.

### 4.2. TopolView Analysis

Various node layout methods are applied to layout the case network nodes in this section. Firstly, different layout methods are compared from three aspects: The display effect of single-layer

structure, the display effect of multi-layer contrast and the computational complexity, and verifying the advantages of multi-force layout method in multiplex networks visualization. Then, the support of the topology view is evaluated for the tasks in Section 3.2.

Figures 11–13 show the layout results of three kinds of layout methods, respectively—the same layout method based on FR model [13], the independent layout method considering the centroid attraction (ZRQ) [14], and the multi-force directed layout method (proposed in Section 3.4). In these figures, sub-figures (a–c) correspond to the original network layers, and sub-figures (d–f) are, respectively, the overlap layer of layer1 and layer2, the overlap layer of layer1, layer2, and layer3, and the aggregation layer.

In Figure 11, the FR layout is applied on the layer1, and the nodes of the other layers are laid out at the same position as the corresponding nodes in the layer1. In other words, each node in the network is located at a same position in all layers. Though this layout facilitates the rapid identification of nodes, nodes belonging to different groups are interleaved in other network layers. As such, it's difficult for users to get the structure components of every network layer, and it is not helpful to further analyze the distribution of nodes and edges.

In contrast, the latter two layout methods do well in showing the community partitioning results in each layer because the basic FR model is used in Figures 12 and 13. The FR model ensures that nodes belonging to the same community are laid out centrally relative to other ones. The nodes within the community are more closely clustered in the independent layout without the attraction from the replica nodes, as shown in Figure 12.

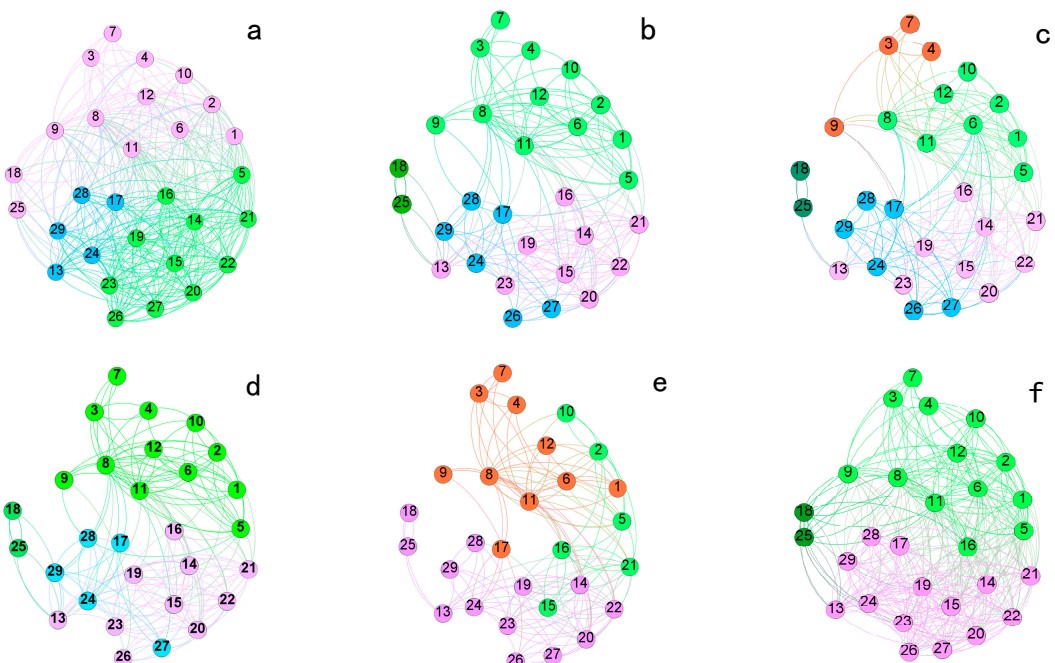

**Figure 11.** Topology views with the same layout method: (**a**) layer1, (**b**) layer2, (**c**) layer3, (**d**) the overlap layer of layer1 and layer2, (**e**) the overlap layer of layer1, layer2, and layer3, and (**f**) the aggregation layer.

However, there are three inadequacies with the independent layout (Figure 12). First, the internal nodes of the same community are excessively aggregated because of the centroid attraction. If the improper parameters are selected, the nodes in the same community will be overlap, such as the community C1 (dark green) in sub-figure (b). Second, the dense nodes have a large degree of occlusion on the internal edges in the community, and it is difficult to clearly display the internal distribution of the communities, such as the community C1 (purple) and the community C2 (blue) in Figure 12a,b. Third, it is also the most critical of multi-layer comparative analysis in that it is difficult to quickly

find and identify the same nodes and similar communities in cross-layer comparison. This is because the nodes of each network layer are laid out separately, and the locations of the community centroids between layers are random. For example, the C3 community is distributed in different positions on the three canvases in Figure 12a–c. Though the three communities have large numbers of overlapping nodes inside, there is a large positional offset between layers.

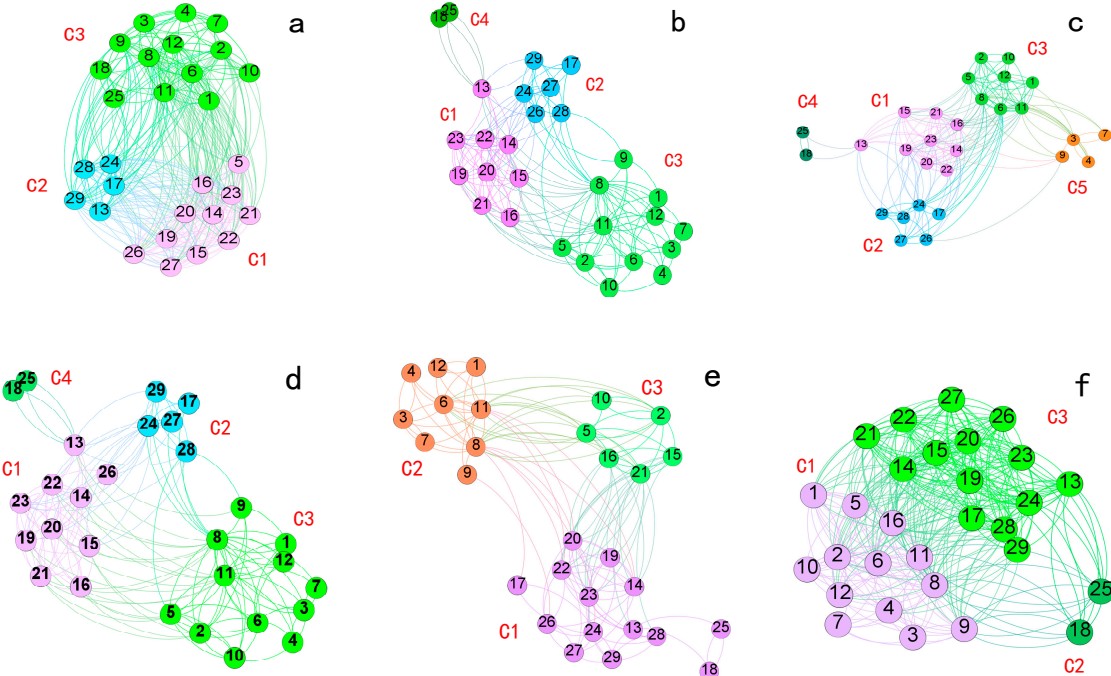

**Figure 12.** Topology views with the independent layout method: (**a**) layer1, (**b**) layer2, (**c**) layer3, (**d**) the overlap layer of layer1 and layer2, (**e**) the overlap layer of layer1, layer2, and layer3, and (**f**) the aggregation layer.

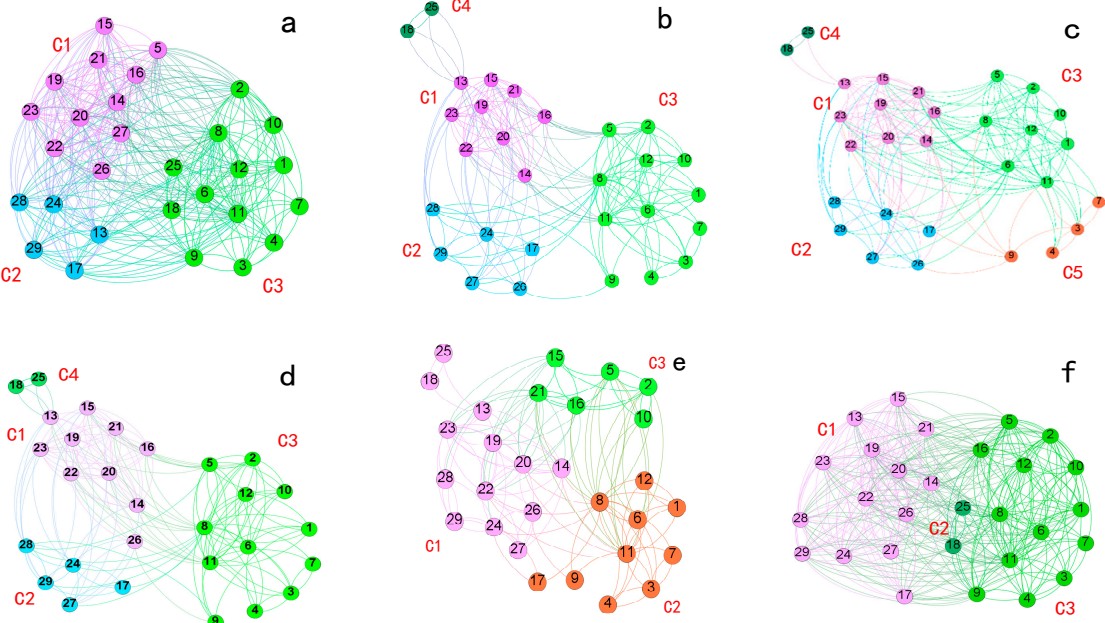

**Figure 13.** Topology views with the multi-force directed layout method: (**a**) layer1, (**b**) layer2, (**c**) layer3, (**d**) the overlap layer of layer1 and layer2, (**e**) the overlap layer of layer1, layer2, and layer3, and (**f**) the aggregation layer.

On the contrary, in the multi-force layout (Figure 13), the internal nodes of the community are gathered to the community center by the centroid attraction, which is helpful to form the obvious community division. On the other hand, the node is prevented from gathering to the community center excessively by introducing the inter-layer attraction between the node and the replica. Simultaneously, the position offsets between the nodes and their replicas in different layers are decreased, such as the node 1,7,3,4 in Figure 13a–f. That is to say, the replica nodes can be found quickly at the corresponding position of the node in other layers. The locations of the communities with more overlapping nodes are relatively fixed in the canvas. For example, all the C1 communities in sub-figures (a)–(f) are placed on the left side of the canvas, and the internal nodes of this community are relatively fixed. In summary, the multi-force layout method facilitates the comparison of the inter-layer structure and the rapid realization of cross-layer identification of similar communities and overlapping nodes.

In terms of computational complexity, the difference is small between the three methods. The computational complexity of the traditional FR model includes the computational complexity of the repulsion $O(|V|^2)$ and the computational complexity of the attraction $O(|intraEdge|)$. The ZRQ model increases the complexity of centroid attraction calculation $O(|comm|)$ based on the FR model. The multi-force model increases the computational complexity of the inter-layer attraction from the counterpart node, based on the ZRQ model. However, the computational complexities of centroid attraction and the cross-layer attraction from node-replica are relatively small compared to the repulsive force. Therefore, for large-scale network nodes layout, all the computational complexity can be regarded as $O(|V|^2)$ in the traditional FR model, the ZRQ model, and the multi-force model. In other words, repulsive force calculation is still the most important factor affecting the computational complexity of the layout process.

Through the comparative analysis of the layout effects of the above three layout algorithms, what can be seen is that the multi-force model can clearly show the community of the network layer in the multiplex network by introducing the centroid attraction and the inter-layer attraction from the counterpart node. Furthermore, the node distribution is more uniform, which can achieve the effect of inter-layer community structure comparison and node-replica identification

The following is a detailed analysis of the support for the multiplex network visual analysis tasks shown in Section 3.2 based on the topology views in Figure 13.

As shown in Figure 13a–f, each network layer (in turn, each original network layer, overlapping layer, full-network overlapping layer, and aggregation layer) is displayed on a two-dimensional plane by using a slice model, which is convenient for users to find and select the region of interest across different layers.

In Figure 13, nodes in different communities are coded in different colors. The nodes in the same communities are coded in the same color, and are clustered together to form a dense layout within the community, while a sparse layout between communities. As shown in sub-figures (a) to (f), the numbers of communities in each network layer are 3, 4, 5, 4, 3, 4 (Task 1.1.1). The network layers are not same in terms of community composition, and the nodes are not completely consistent with the same community (Task 2.1.1). As shown in Figure 13b,d, the community partitions in the two network layers are basically the same, and the positions of the nodes in the canvas are relatively fixed. However, it can be clearly seen that node 26 in Figure 13d is far from the community C2 (blue) and close to community C1 (purple). This is because the edges distributions of the two network layers are different, and the community divisions of the nodes 26 in the two network layers changes, resulting in the effect of the node 26 approaching to community C1 (purple) in the Figure 13d. Similarly, comparing Figure 13a with Figure 13b, it is very intuitive to see that the C1, C2, and C3 communities have a large proportion of overlapping nodes, but there are still many nodes distributed in different communities. In short, the adopting of multi-force layout is convenient to identify overlapping nodes and non-overlapping nodes in similar communities.

Furthermore, a single network layer can be focused on to analysis the distribution of nodes within the community, the distribution of edges within the community, and the distribution of the edges between the communities. Next, take the Figure 13b as an example for analysis. In terms of the community composition of the network layer, this layer consists of four communities (coded in dark green, purple, light green, and sky blue, respectively), and the C4 community (dark green) has the fewest number of nodes (two nodes), and the C3 community (light green) contains the most nodes (12 nodes). (Task 1.1.1). In the distribution of edges within the communities, the C1 community (purple) and the C3 community (light green) have more inner-edges, while the other two communities are relatively sparse. In the aspect of the distribution of edges between the communities, the C4 community (dark green) is only connected with the C1 community (purple), while edges between the other three communities are evenly distributed with each other (Task 1.1.2). Based on the analysis of each network layer, it is also possible to compare the distribution of nodes and edges within similar communities between multiple network layers.

From the intuitive analysis of the community composition, nodes distribution, and edges distribution, we find that the topology views can partially solve Task 1.1.1 and Task 1.1.2 in Section 3.2., and it is also possible to qualitatively compare the distribution of communities, nodes and edges between layers (Task 2.1.1, Task 2.2.1 and Task 2.2.2). Though it is intuitively to aware that there is a large degree of overlap between the network layers, it is difficult to give quantitative comparison results such as the number of overlapping edges or the number of overlapping edges only via the topology view. Therefore, it is difficult to meet the needs of large-scale network analysis or more accurate information. However, this problem can be solved by the combination of interaction method and high-level infographic-style view proposed in this paper and the specific analysis is introduced in Section 4.3.

As the end of this section, the role of the topology view in the multiplex network analysis can be summarized in the following three aspects:

1. Displaying the details of the nodes and edges in the network layer through the node-link diagrams, revealing the community composition in every network layer.
2. Assisting users in perceiving the difference of the inter-layer structure on the intuitive, evaluating and predicting the distribution of the nodes and the edges in network, which facilitate users to find the layer (or sub-graph, region, etc.) and nodes of interest to further explore their interests.
3. Supporting selection between multiple layout algorithms, which can be used to carry out deeper research on node automatic layout algorithm for multiplex network.

*4.3. Similarity Pattern Analysis*

This section shows the way similar pattern representation support the multi-layer comparative analysis tasks described in Section 3.2.

Based on the topology views of Section 4.2, high-level patterns representations are drawn in the high-level infographic-style view after selecting the layer of interest (or sub-graph, region, etc.) and aggregating nodes and edges information by selecting and filtering components.

First, in order to explore the degree of overlap of nodes and edges between the network layers, the three original network layers are chosen to make a comparative analysis. The distribution of overlapping edges is shown in Figure 14. The three circles respectively represent the edges sets in each network layer encoded by different colors. The intersection areas represent the edges that are repeated in two or three network layers, while the sizes of the area are proportional to the numbers of the overlapping edges.

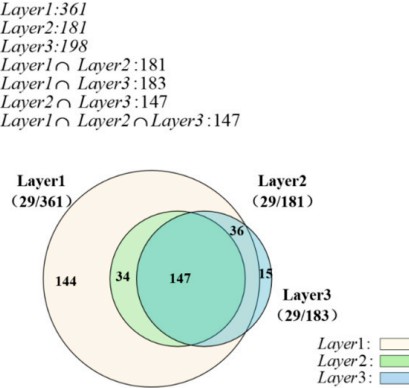

**Figure 14.** Distribution of overlapping edges in the three original layers.

From Figure 14, we can get the number of nodes and the number of edges in each network layer from the values in parentheses, with the format of "(nodes-number/edges-number)". Each layer contains all the nodes in the network (Task 1.2). Among the three circles, layer1 gets the largest size, which means that layer1 has the largest number of edges, namely 361, followed by layer3 with 198 edges (Task 1.3). In addition, the similarity of edges between these layers can also be analyzed (Task 2.1.3) as follows: In Figure 14, layer2 contains 181 edges which are completely included in the layer1, indicating that the two layers have the highest degree of similarity. While, the overlap area of the layer3 and the layer1 is relatively small, but it also exceeds half the area of layer3, with the number of 147. The number of the overlapping edges of all the three original network layers is 147, showing that the connection relationships represented by the three network layers are largely repeated. layer3 has a small number of special connection relationships that are not in the layer1.

By analyzing the distribution of the overlapping edges in the three original network layers in Figure 14, we can find that layer 1 contains the most edges. To explore the distribution characteristics of nodes and edges in layer1, the aggregation layer and the overlap layer (Task 2.1.2 and Task 2.1.3), are selected to create a similarity pattern representation, as shown in Figure 15. The edges in layer1 account for 96% of all edges in the aggregation layer, suggesting that the connections in layer1 basically cover all the connections of nodes in all network layers.

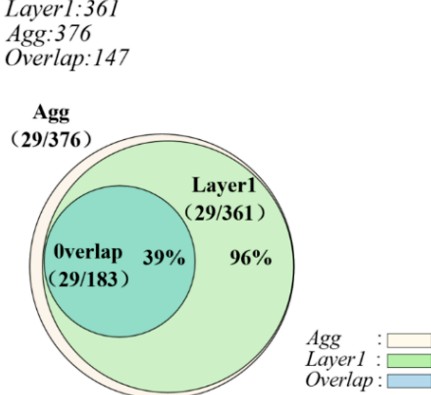

**Figure 15.** Distribution of overlapping edges between layer1, the aggregation layer and the overlap layer.

However, there are still a few of the edges in the aggregation layer that are not in layer1. As shown in Figure 12, this part of the edge is mainly from the layer3. The edges in the overlap layer represent the connections shared in the three network layers, which reveals the core network structure of the network data, to some extent. It can be seen from the figure that the edges in overlap layer occupies 39% of the aggregation layer. Since the three network layers in the case network contain all the network nodes, the similarity of the nodes is not separately analyzed.

Then, we select the corresponding similar communities in the original network layer and analyze the number and distribution of overlapping nodes and overlapping edges in the specified areas (Task 2.2). As shown in Figure 16, three areas are correspondingly selected in the three original layers, which are area1 (purple), area2 (blue), and area3 (green). The selected elements are the nodes and all the edges of the nodes inside the box.

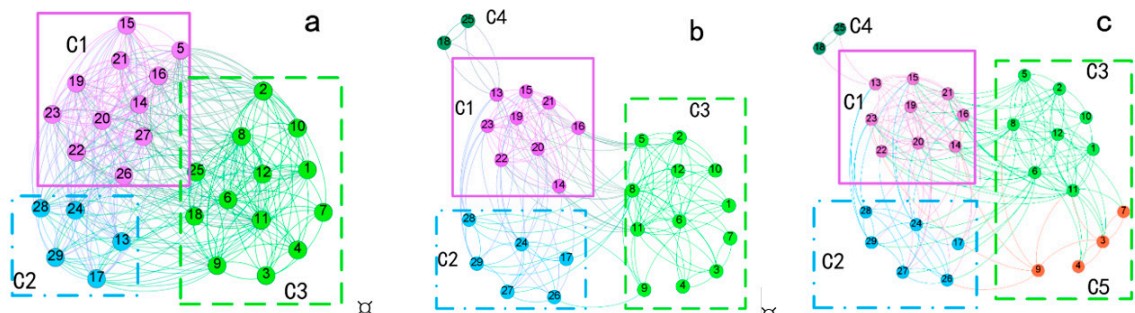

**Figure 16.** Topology views with areas selected (**a**) layer1, (**b**) layer2, and (**c**) layer3: The corresponding three areas are area1 (purple), area2 (blue), and area3 (green).

Figure 17 shows the distribution of overlapping nodes in the similar areas in the three original network layers. The layer labels layer1, layer2, and layer3 represent the given areas in the corresponding network layer. When the number of nodes is small, the user can directly obtain the number of nodes from the topology view. For example, the number of nodes included in area2 of the three network layers is five in Figure 12. In addition, it is necessary to compare nodes one by one to answer the following questions, "which one is the overlapping node," "which is the self-owned node," and "what is the number of overlapping nodes." However, the method above is inefficient. Specially, when the number is slightly larger (e.g., there are more than 10 nodes), it is difficult for users to effectively obtain the number of nodes in the area and the distribution of the overlapping nodes.

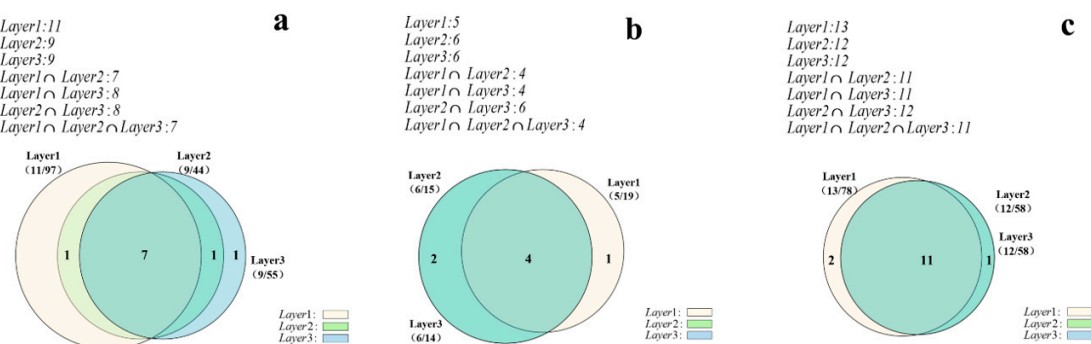

**Figure 17.** Distribution of overlapping nodes in selected areas: (**a**) area1, (**b**) area2, and (**c**) area3.

In contrast, the area-proportional Venn diagrams is used to present the number of nodes in the corresponding area and the number of overlapping nodes in the three network layers (Task 2.2.1), as shown in Figure 17. The node repetition rate is high in the corresponding areas of the three network layers. The number of overlapping nodes in the three network layers in the three areas is seven, four, and 11, respectively, accounting for more than half of the total number of internal nodes in the given areas of each network layer. The area2 in layer2 and the layer3 are completely overlapping, as well as the area3 in layer2 and the layer3.

Figure 18 shows the distribution of overlapping edges of the specified areas in the three network layers. From this figure, the number of the overlapping edges and the number of the own edges in the three network layers can be analyzed (Task 2.2.2). Taking area1 as an example shown in sub-figure (a): (1) The sum of the internal edges of the specified areas of the three network layers is 97, 44, and 55,

respectively, and the layer1 has the most edges; (2) the number of the overlapping edges of the three network layers is 34, which is more than 60% of the number of edges in the specified area of the layer2 and the layer3; (3) the number of the overlapping edges in the layer 2 and layer 3 is 39, indicating that the structures are quite similar in layer 2 and layer 3.

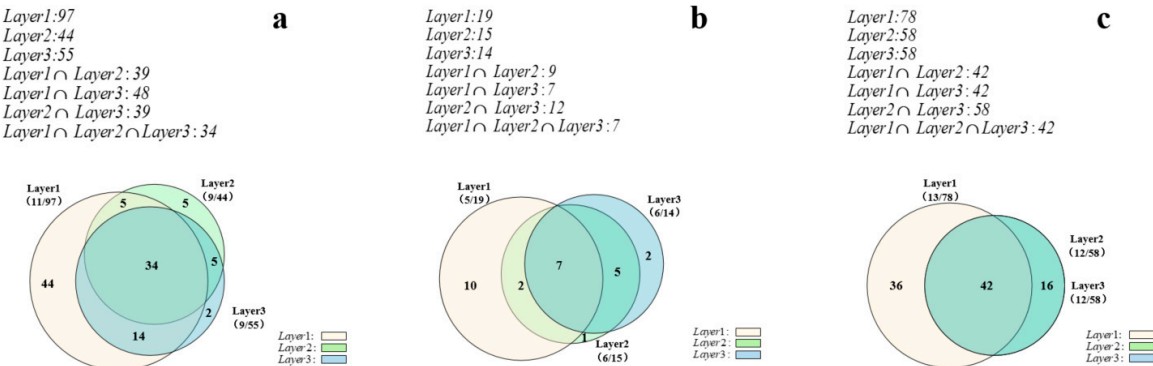

**Figure 18.** Distribution of overlapping edges in selected areas: (**a**) area1, (**b**) area2, and (**c**) area3.

In summary, the above-mentioned area-proportional Venn diagrams can be used to visually represent and analyze the high-level patterns information such as the number and the distribution of overlapping nodes and edges in the specified areas and can be used as a supplement to the topology view in quantitative analysis of interlayer differences (or similarity). The analysis results show that this method can efficiently support the inter-layer comparative analysis tasks in Section 3.2 (Task 2).

*4.4. Interaction Pattern Analysis*

This section shows how does the similar pattern representation support the single-layer network analysis tasks in Section 3.2.

First, based on the analysis and comparison of all network layers in Section 4.3, the network layer of interest is determined for key analysis (here the layer with the largest number of edges is selected as an example). Then, the areas of interest (or community) are created on the topology view named as group1, group2, and group3, and the interaction pattern representations are generated after the attribute information of the nodes and the edges is counted on the backend. Finally, the interaction pattern is uncovered by analyzing the regularity of the change of the number of edges in and between the groups in the given layer.

Figure 19 shows the topology views of the layer1. Three groups are selected and the nodes are filtered by the degree to obtain the topology views of the network on the four intervals, i.e., (a) $Degree \in [8, 42]$, (b) $Degree \in [18, 42]$, (c) $Degree \in [28, 42]$, and (d) $Degree \in [35, 42]$, corresponding to the four sub-figures. From the changes of topology, we can see that: The degree values of all nodes in layer1 are generally large, the minimum value is 8, and the maximum value is 42 (since the network is a directed graph, the maximum value of degree exceeds N-1); When the minimum value is increased to 18 (shown in Figure 19b), the numbers of nodes are not reduced in the group1 and group2, while the number of nodes in the group3 is reduced by six; when the minimum value continues to increase to 28, only one node remains in group2 (node 28), implying that the range of node degrees in the group2 is mostly from 18 to 28; when the minimum value is increased to 35, only one node in group3 and two nodes in group1 remain in the network, indicating that these three nodes are the ones with the highest degree value in this layer.

Based on the above analysis of the changes in the topology views, we can roughly obtain the distribution of the nodes in the communities. As the degree increases, we find that the number of edges within and between the communities is reduced. However, these changes cannot be revealed by numerical comparisons.

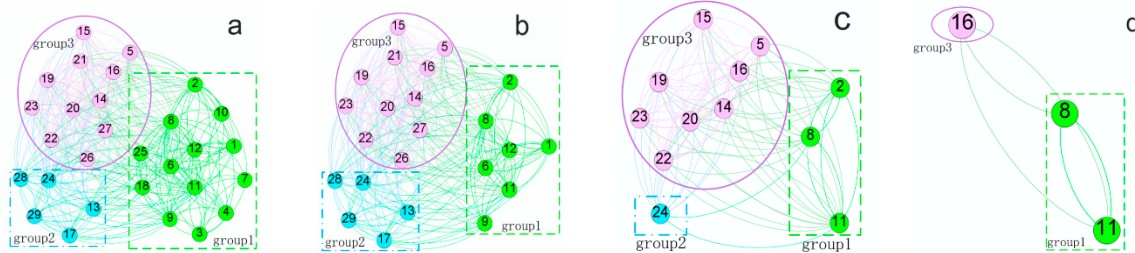

**Figure 19.** Topology views of layer1 at different ranges of degree: (**a**) *Degree* ∈ [8, 42], (**b**) *Degree* ∈ [18, 42], (**c**) *Degree* ∈ [28, 42], (**d**) *Degree* ∈ [35, 42].

Figure 20 shows these changes of the interaction pattern inside and between the groups corresponding to the four sub-figures in Figure 19, respectively. There are three box containers in the Figure 20, representing the three groups in Figure 19a, respectively, and the internal values of the box containers are the number of nodes. If necessary, the degree distribution of the nodes can also be displayed in the containers, as well as the various attribute visualization charts such as PageRank sorting and centrality sorting, as shown in Figure 3c.

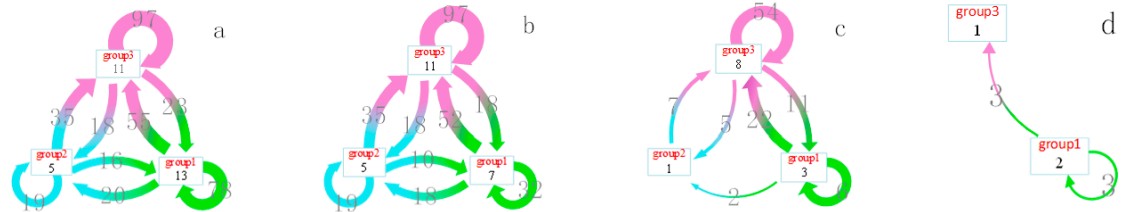

**Figure 20.** Interaction pattern representations of layer1 at different ranges of degree: (**a**) *Degree* ∈ [8, 42], (**b**) *Degree* ∈ [18, 42], (**c**) *Degree* ∈ [28, 42], (**d**) *Degree* ∈ [35, 42].

From Figure 20, the number of internal nodes in the community (Task 1.1.1), the distribution of edges in and between the communities (Task 1.1.2) can be obtained more intuitively and accurately. Taking Figure 20a as an example, we can get the following conclusions. The numbers of nodes in group1, group2, and group3 are, respectively, 13, five, and 11, and the group1 in Figure 20a gets the most nodes. The self-circulating arrows indicate the internal edges of the communities, which are 78, 19, and 97, respectively. The internal edges in group3 are the most, indicating that nodes within this community interacts most frequently with each other. The directed arrows between the box containers represent the interactions of nodes between communities. The edges between group3 and group1 is the most, and the number of edges of group1 and group2 pointing to group3 (35 and 55, respectively) are significantly more than that of group3 pointing to the other two groups. That is to say that the members of group3 actively contact more with the members within the group1 and group2.

As shown in Figure 20b, when the minimum value is increased to 18, the number of nodes inside group1 is reduced by six, the internal edges and the edges pointing to the other two groups are reduced more, and the number of the edges inside group2 and group3 and the edges pointing to these two groups do not change. In other words, the nodes that are filtered out in the group1 (such as nodes 10, 4, and 3) have the connections with the internal members within the local group but not with the internal members of the group2 and group3. In Figure 20c, when the minimum value continues to increase to 28, most of the nodes in the three groups are hidden, only one node is left in group2, and the number of edges between the groups is reduced more. There are only three edges in group1 and three edges pointing to group3 from group1 when the minimum value is increased to 35, as shown in Figure 20d.

Similarly, the layer2 and the layer3 are respectively selected, the groups of interest are set according to the community division of each network layer, and the interaction patterns between the communities in each layer are plotted, as shown in Figures 21 and 22. The following basic information can be obtained from the figures. There are four communities in layer2 and five in layer3, and the distribution

of nodes can also be obtained in each community. In the terms of the distribution of edges, group1 and group3 in layer2 have the most internal edges, as well as group3 in layer3. However, group1 in layer2 and group3 in layer3 have more interactions with other groups, and only group1 has edges pointing to group3 in layer2 and layer3. It is indicated that in the two network layers, group1 actively contacts group3 and vice versa.

In addition, the similarity pattern can also be comparative analyzed based on the interaction pattern described above. By comparing Figures 20a, 21 and 22, the difference among these three network layers can also be obtained in community composition, nodes distribution, edges distribution in and between communities.

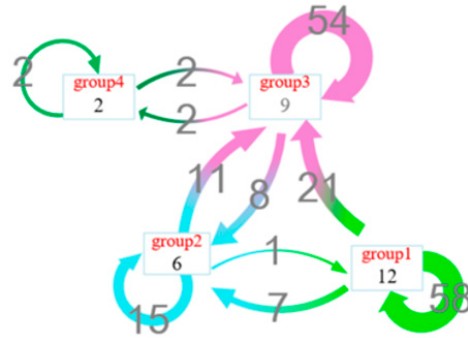

**Figure 21.** Interaction pattern representation of layer2.

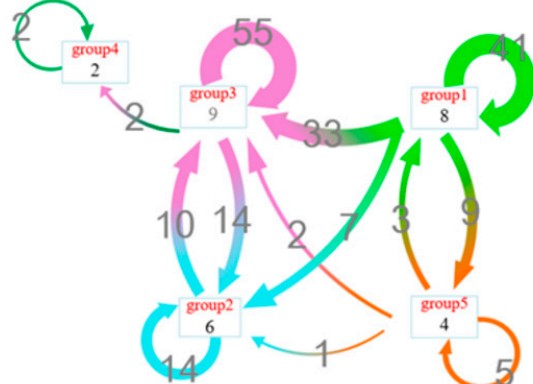

**Figure 22.** Interaction pattern representation of layer3.

In summary, the method is effective by using directed arrows to represent the internal and inter-regions aggregation information of the selected regions, which can more intuitively and accurately display the community composition of the network layer and the edges distribution in and between the regions. Such information effectively assists in the exploration of interaction pattern between regions. By combining the node attribute filtering to hide some nodes that are not of interest, this method can also be used to explore the changes of the interaction pattern between the network layers more accurately. In addition, the directed arrows can efficaciously indicate the direction of the edges between the regions, avoiding the fuzziness of the edge direction in the topology view.

## 5. Discussions and Limitations

### 5.1. Discussions

The motivation of this paper is to visually and quantitatively compare the structure of multiple network layers in detail and overview when studying multiplex networks. Then, a generalized approach is provided to assist in the analysis and exploration of network data with multiple types of connection relationships, such as multi-platform social networks. The basic idea is relatively

simple: (1) Combines topology details with overview information, (2) selects the layer of interest (or sub-graph, region, etc.) by simple, direct selection and filtering, and (3) presents information to users in a user-familiar metaphor.

At the beginning of the design, the prototype only includes the topology view, and it is hoped that the comparison analysis of the multi-layer structure can be realized only through the node layout in the topology view. However, it is difficult to provide users with more intuitive and in-depth knowledge only through the node-link diagrams. That is because the conclusions obtained from visual perception should be always confirmed by the numerical information, both for expert users and non-expert users.

Based on different characteristics of the detail view and the outline view, this paper develops a multi-force directed model and two high-level patterns representations. The former is used to display the visual contrast of the interlayer structure, and the latter is used to show the numerical comparison of the patterns. Some simple and direct interaction are also designed to help users freely identify, select, and compare the layers of interest (or sub-graphs, regions, etc.). The experiment shows that the method proposed can fully support the multiplex networks analysis tasks described in Section 3.2.

In Table 2, we use ★ to express the support for analysis tasks, and a greater number of ★ indicates a strongly support for analysis tasks. In Table 2, we use the number of the symbol ★ to identify how different platforms support the single-layer visual analysis and multi-layer visual analysis. Different methods proposed in the software or prototypes related to the multiplex networks visualization are mainly compared from layout algorithm (LA), high-level infographic-style presentation (HIP), and interaction to the support for the single-layer visual analysis and multi-layer visual analysis.

**Table 2.** Comparison of analytical capabilities of different multiplex networks visualization platforms.

| Name | Single-Layer Visual Analysis | | | Multi-Layer Visual Analysis | | |
|---|---|---|---|---|---|---|
| | LA | HIP | Interaction | LA | HIP | Interaction |
| Gephi [36] | ★★★★★ | ★★★★ | ★★★★★ | ★ | ★ | ★ |
| Multired [3] | ★★★ | ★ | ★ | ★★ | ★ | ★ |
| Multinets [7] | ★★★ | ★ | ★ | ★★ | ★ | ★ |
| MuxViz9 | ★★★ | ★★★ | ★★★ | ★★ | ★★ | ★★★ |
| Prototype proposed | ★★★ | ★★★ | ★★★★ | ★★★ | ★★★ | ★★★ |

In terms of single-layer analysis, Gephi is a mature graph visualization software. That is because Gephi integrates rich node layout methods, node and edge editing methods, and multi-attribute scatter rendering methods. This paper draws on the designing ideas from Gephi, integrating some classic node layout algorithms and developing an interactive interface that supports node filtering and region selection—specifically, the interaction pattern representation for single-layer visual analysis. In contrast, Pynet, Multinetx, and MuxViz are specialized prototypes for multi-layer network analysis that are are relatively weak in supporting the single-layer visual analysis.

Aiming at improving multi-layer visual analysis, Pynet, Multinetx, and MuxViz use a cascading slice model to display multiple layers in a 2.5D view, enabling multi-layer network node layout methods, but these three prototypes are short of the node layout algorithm specifically for multi-layer networks. Particularly, Pynet and Multinetx are mainly designed for network analysis, and do not support the generation of high-level infographic-style views and friendly interaction. As such, they are difficult for non-expert users to operate in the field of network visualization. MuxViz provides some visual charts only for attribute comparative analysis in multi-layer networks, but there is a lack of region selection and the support for multi-regional comparison between layers. In contrast, a kind of prototype is designed in this paper—one with a multi-force model layout algorithm, two high-level patterns representations, and necessary selection and filtering operations—which is superior to other software and prototypes in supporting multi-layer comparative analysis.

Finally, the proposed prototype is based on the B/S architecture and is implemented in pure JavaScript language for easy deployment and use on a web browser.

*5.2. Limitations and Future Directions*

Though the implementations in this paper well support the multiplex networks layer-edge patterns analysis tasks,9 we believe that the research still has the following shortcomings and improvement directions combined with user needs in comparative analysis of related research):

1.  This paper mainly provides an automatic layout method specially for multiplex networks visualization. However, the computational complexity of repulsion calculation is still large, and it is time-consuming to lay out large-scale nodes. The novel layout methods should take quadtrees, multidimensional scaling analysis, and many other methods into consideration to speed up node position calculation. Some other kinds of community detection algorithms can also be applied to effectively support the comparison analysis of the multi-layer structure.
2.  In terms of similarity pattern representation, the area-proportional circular Venn diagrams do better in visual perception. However, when the number of input sets is larger than six, the layout and size calculations will not be accurate. It is not suitable for the representation of the relationship between a larger number of sets. Since the specific analysis focuses more on the two- or three-layer comparison, the method in this paper basically meets the requirements.
3.  This article provides the necessary selection and filtering methods to basically satisfy users' freedom to select the node elements of the specified area. However, the region selection requires community partitioning and force-directed layout in advance to cluster nodes that belong to a same community. Considering the scalability of the system, there is still a need for multiple styles of interaction to support freewill exploration.

**6. Conclusions**

According to the requirements of multiplex networks layer-edge patterns exploration and analysis, this paper proposes an interactive method for the exploration and analysis that closely couples topological structure and high-level patterns information through selection and aggregation. This method can compare the multi-layer structure visually and specially quantitatively through the topological view and the high-level informatic-style view created in this paper. This method can also ensure users to focus on the region of interest through selection and filtering and continuously guide the user to deepen the understanding of the multi-layer network structure.

In the experimental part, the effectiveness of the proposed method is verified on real-world data. The analysis results show that the approach proposed can well support the multiplex networks layer-edge patterns analysis tasks. This method fully embodies two main functions for visual analysis in multiplex networks: On the one hand, it can be treated as a kind of exploration tool to assist users to obtain relevant revelation of network structure or attribute; on the other hand, it can be used to display the results of pre-existence or excavation in an understandable way.

Finally, this paper compares the proposed method with some other related researches and analyzes the differences in single-layer analysis and multi-layer comparative analysis, pointing out the limitations in analyzing large-scale network data, as well as the direction of improvement.

**Author Contributions:** Conceptualization, X.Z.; formal analysis, X.Z.; funding acquisition, L.W.; project administration, L.W.; resources, L.W.; supervision, S.Y.; validation, S.Y.; Writing—Original Draft, X.Z.; Writing—Review and Editing, K.L.

**Funding:** This work was supported by the Equipment Pre-Research Foundation under Grant No. 6142010010301.

**Conflicts of Interest:** The authors declare no conflict of interest.

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
