# Peer review of "Layer-Edge Patterns Exploration and Presentation in Multiplex Networks: From Detail to Overview via Selections and Aggregations"

_electronics, doi:10.3390/electronics8040387_

Round 1

Reviewer 1 Report

This manuscript presents an interesting and potentially useful approach to visualizing multiplex networks, is relatively well written and it thus can be considered for publication in Electronics. Before my final recommendation I would as the Authors to address the following question: Is it possible to use the proposed layout algorithm to create a single layout, somehow "optimal" for all the layers, and shared by all of them, that would fix nodes' positions? It would be similar to the "same layout" method, but taking into account the multilayer nature of the network. Would it be viable? Some corrections are also needed as those related to the size of figures, notation inconsistency and spell-checking. The corrections needed are indicated directly in the attached pdf of the orginal submission.

Author Response

Response to Reviewer 1 Comments

Point 1: Is it possible to use the proposed layout algorithm to create a single layout, somehow "optimal" for all the layers, and shared by all of them, that would fix nodes' positions? It would be similar to the "same layout" method, but taking into account the multilayer nature of the network. Would it be viable?

Response 1:

Thanks for your valuable suggestions for our manuscript.

We have carefully considered and experimented with the idea about the application of the proposed layout algorithm. Unfortunately, we didn't get better results. First of all, each network layer should highlight its own structural characteristics. Because the connections between nodes in each layer are different, the force states of the nodes in different layers are diverse. So, it is difficult to achieve an optimal layout effect through automatic layout, where nodes in all network layers have the same location, while showing the structural characteristics of each layer. Then, we tried to layout the nodes with the positions of nodes in aggregation layer or the overlapping layer, so that the nodes in each network layer have the same layout position, and then draw the edge according to the connections of each layer respectively. However, the resulting node locations still do not highlight the structural characteristics of the individual network layers themselves. In summary, the optimal layout is difficult to achieve, and it will provide motivation for further research on multiplex network visualization.

Point 2: Some corrections are also needed as those related to the size of figures, notation inconsistency and spell-checking.

Response 2:

Related errors have been corrected, please refer to the manuscript for details. Here is the list of all the corrections:

Line 87: Modified grammar problem, with “are” instead of “is”.

Line 140: Modified grammar problem, with adding “can be”.

Line 238: Improved the description of the review of Louvain algorithm.

Line 255: The figure has been zoomed to make it to be clear.

Line 257-259: Modified grammar problem, by changing the sentence with a clearer description language.

Line 260-261: Explained the difference between the notations of D, Ka, Kr in traditional FR model with the notations of  ,  , ,  in multi-force model.

Line 273: Improved the description of the notation of  , which is necessary to help understand the formula (4).

Line 352: Figure7 has been zoomed to make it to be clearer.

Line 395-400: Refined the description and analysis of the Figure8.

Line 415: Refined the title of Figure 9.

Line 486-492: Refined the description and analysis of sub-figures (a)~(f) of Figure 13, which prove the conclusion more directly.

Line 506-508: Refined the description of how does the repulsive force calculation affect the computational complexity of the layout process.

Line 665: “quantified analysis” is instead by “revealed by numerical comparisons”, the latter is more easily to be understood.

Line 718: “generally method” is instead by “generalized approach”. The approach proposed in this paper is a kind of generic approach, which can be used in many multi-plex network datasets, such as society, transportation, biology, electricity and public infrastructure.

Line 726-728: Improved the description of the disadvantage of node-link diagrams.

Line 736-740: Improved the description of table2. Refined the analysis of different column.

Line 750: Modified grammar problem, with “Aiming at improving …” instead of “Aim at…”.

Lin 753-756: Modified grammar problem, improved the description of the limitions of previous platforms.

Lin 772-773: Improved the description of future direction about the application of different community detection methods.

Reviewer 2 Report

This paper deals with the exploratory analysis and visualization of multiplex networks. The paper is well-written but it sounds very technical.

The main remark that I can make is that they use the Louvain method for community detection claiming that this method does not fail to detect small communities (line 239). Multiple studies highlighted the fact that modularity, the quality measure used by Louvain algorithm, suffers from resolution limit (see for instance [a], [b]). Indeed, modularity is largely used for community detection and it will work properly for small networks, like the one the authors studied in section 4 (only 29 nodes and 740 edges). However, I suggest the authors to apply the proposed method to larger networks and see what happens. They can also see the results with other quality functions for which the Louvain method works properly (see for instance, [c])

[a] Fortunato, S. and Barthelemy, M. (2006). Resolution limit in community detection. In Proceedings of the National Academy of Sciences of the United States of America

[b] P. Conde-Céspedes, F. Marcotorchino and      E. Viennet, "Comparison      of linear modularization criteria using the relational formalism, an      approach to easily identify resolution limit", AKDM6,      "Advances in Knowledge Discovery and Management", pp. 101-120,      2017.

[c] R. Campigotto, P. Conde-Céspedes and      J-L Guillaume, A Generalized and Adaptive Method for Community Detection.

Additional remarks:

Line 52 : How DO the nodes or edges overlap between two different layers? 

Line 137: … and nodes on different layers can be connected to each other only in the same layer : I found this difficult to understand. Do you mean nodes can be connected only in the same layer? If possible, simplify this sentences

Author Response

Response to Reviewer 1 Comments

Point 1: The main remark that I can make is that they use the Louvain method for community detection claiming that this method does not fail to detect small communities (line 239). Multiple studies highlighted the fact that modularity, the quality measure used by Louvain algorithm, suffers from resolution limit (see for instance [a], [b]). Indeed, modularity is largely used for community detection and it will work properly for small networks, like the one the authors studied in section 4 (only 29 nodes and 740 edges). However, I suggest the authors to apply the proposed method to larger networks and see what happens. They can also see the results with other quality functions for which the Louvain method works properly (see for instance, [c])

Response 1:

Thanks for your valuable suggestions for our manuscript, especially for the community detection method. According to the commends, we downloaded and seriously studied the related literature. What we found is that the Louvain method is not really perfect, through it has been widely used in community detection. And there are so many modified versions to make an improvement of the time efficiency or the quality of the partition result.

Firstly, quality functions such as modularity really suffers from resolution limit, which has been fully verified in ref [a, c]. Specifically, community detection methods based on quality functions, i.e., such that the quality of a partition is given by the sum of the qualities of the individual modules, tend to get a smaller number of modules, and some of these modules are the groups of smaller partitions. Then, a refined partition can not be identified, resulting in the missing of important substructures of a network. So that, a new theoretical framework is really needed that focuses on a local definition of community, rather than on definitions relying on a global null model.

Secondly, regarding the issue above, a generic version of Louvain algorithm is proposed in ref[b], resulting in an optimal partition of the criteria with real networks of different sizes. Ref[b] provided a very in-depth research on community detection, and proposed the relational notation to identify the criteria which is suffering from the resolution limit from a theoretical point of view. In ref [c], any linear criteria and several non-linear criteria have been integrated efficiently to the generic version of Louvain to approach the optional solution. So, both the ref [b] and ref [c] have carried out in-depth research on Louvain method and are of great significance in the field of community detection, as well as the researches of network analysis and visualization.

The main work of this paper is to provide a generalized approach to assist in the analysis and exploration of multi-plex network data based on the detection of community structure. Users can analyze the mesoscale structure of the networks from different granularities or views, by freely choosing different community detection methods. To ensure the paper’ accuracy, we have deleted the inappropriate description and added an introduction to the research results of ref a, b and c in the section “3.4.1. Community Detection”, as well as other researches about Louvain algorithm. Without loss of generality, the traditional Louvain algorithm is introduced as a representative community detection algorithm. And related references are added in the section “references”, with the number 29-33. In the outlook section, the application of multi-type community detection algorithms has been added as one of the main research directions in the future. In addition, we will further study the ref [a, b, c] and some other related literatures, and continue to pay attention to the impact of community detection resolution in subsequent topics related to network analysis and visualization research.

Point 2: Additional remarks:

Line 52: How DO the nodes or edges overlap between two different layers?

Response: In this paper, the “overlap” means the same node or edge appears in different layers. Then, the number of overlapping nodes or edges may reflect the similarity characters in different network layer. In addition, the degree of difference or similarity between two similar community can be displayed with the number of overlapping nodes and edges, just as what the similarity pattern representation shows.

Line 137: … and nodes on different layers can be connected to each other only in the same layer: I found this difficult to understand. Do you mean nodes can be connected only in the same layer? If possible, simplify this sentence.

Response:

Thank you for the advice, this sentence has been revised in accordance with the expert’s opinion, which is formally what I want to describe.